# Combining benzalkonium chloride addition with filtration to inhibit dissolved inorganic carbon alteration during the preservation of water sample in radiocarbon analysis

Hiroshi A. Takahashi[1], Masayo Minami[2]

[1]Geologidcal Survey of Japan, National Institute of Advanced Industrial Science and Technology, Tsukuba, 305-8567, Japan
[2]Institute for Space–Earth Environmental Research, Nagoya University, Nagoya 464-8601, Japan

*Correspondence to*: Hiroshi A. Takahashi (h.a.takahashi@aist.go.jp)

**Abstract.** Benzalkonium chloride (BAC) addition has shown great promise as a disinfectant for measuring $\delta^{13}C$ and $^{14}C$ of dissolved inorganic carbon (DIC) in freshwater samples. However, it was reported that the effectiveness of BAC to prevent DIC change was reduced for the use of seawater samples. The present study aimed to evaluate the effectiveness of adding BAC as a disinfectant in carbon isotopic analyses of DIC in water samples. We compared the efficacy of BAC addition, filtration (0.22 μm PTFE or 0.2–0.45 μm PES filters), and a combination of BAC addition and filtration in preventing DIC alterations caused by biological activity using the freshwater (salinity <0.5) and the brackish water (salinity ~200) samples. The freshwater sample treated with BAC showed the no alteration of DIC. In contrast, for seawater sample, BAC addition alone did not prevent changes in DIC, but the combined treatment was effective. The $^{14}C$ concentration of samples treated with both BAC addition and filtration exhibited minimal changes, ranging from 0.2–0.4 percent Modern Carbon (pMC) over 41 weeks, despite the addition of sugar included to increase DIC changes several-fold. Although the complete elimination of biological effects may be challenging with the combined method, the observed changes remained within practical limits. Concerns about $CO_2$ contamination during sample filtration were also addressed and found to be negligible. These results suggest that combining filtration and BAC addition is an effective method for suppressing biological DIC alterations in $^{14}C$ analysis, even in seawater samples.

## 1 Introduction

Radiocarbon ($^{14}C$) analysis of dissolved inorganic carbon (DIC) in seawater plays a vital role in the elucidation of seawater circulation and atmospheric-ocean $CO_2$ exchange (Matsumoto, 2007; Mcnichol et al., 2022). For global understanding of ocean water behaviors, it is necessary to analyze samples from various regions over long timeframes, ensuring high-quality and consistent analysis is crucial for maintaining data integrity and comparability across different oceans (Key et al., 2002; Anderson, 2020; Olsen et al., 2020). Standard Operating Procedures (SOPs) for the analysis of seawater have been developed to define the protocols and analytical methods necessary to meet these requirements (Dickson et al., 2007; Abrams, 2013). The SOPs recommend that water samples collected for $CO_2$-related analyses such as DIC, total alkalinity, and $CO_2$ fugacity be treated with a mercuric chloride ($HgCl_2$) solution to prevent biological activity that may alter the carbon distribution in the sample container before analysis. However, the ecological toxicity of $HgCl_2$ poses significant challenges. Additionally, the use of $HgCl_2$ in water samples can lead to uncertainties in the analytical results, as mercury interacts with dissolved organic matter in the water, forming complexes that reduce the total alkalinity, potentially complicating the analysis (Mos et al., 2021). Argentino et al. (2023) reported alternations in DIC concentration and $\delta^{13}C$ values in marine pore water samples from methane seepage areas treated with $HgCl_2$. Given these environmental and practical concerns, alternative preservation methods that avoid the use of mercury are increasingly desirable.

The methods proposed for the preservation of water samples without the use of $HgCl_2$ include refrigeration, filtration, and the addition of non-toxic or less toxic preservatives (Aucour et al., 1999; Doctor et al., 2008; Ascough et al., 2010; Takahashi et

al., 2019b; Wilson et al., 2020; Mos et al., 2021; Takahashi and Minami, 2022). Chemical sterilization methods have been explored, such as adding acids or alkalis to prevent microbial activity in samples intended for the analysis of gases other than $CO_2$, such as methane (Magen et al., 2014). However, altering the pH of water samples is not suitable for DIC analysis, as DIC concentrations are highly sensitive to pH changes.

The addition of benzalkonium chloride (BAC) has shown great promise as a disinfectant for measuring $\delta^{13}C$ and $^{14}C$ of DIC in freshwater samples (Takahashi and Minami, 2022; Takahashi et al., 2019b), and for dissolved $CH_4$ concentrations in swamp water (Osaka et al., 2024). BAC is one of the quaternary ammonium compounds (QACs), a major product of cationic surfactants, and is widely used as a disinfectant (Kuo, 1998; Mcdonnell and Russell, 1999). QACs penetrate cell membranes and disrupt both the physical and biochemical properties of cells (Gilbert and Moore, 2005; Wessels and Ingmer, 2013). As most of the bioavailable fraction of QAC in environmental waters can be reduced by sewage treatment plants (Deleo et al., 2020), the ecotoxicological hazard posed by QACs is far lower than that of mercury. Takahashi et al. (2019b) investigated alterations in DIC concentrations and $\delta^{13}C$ values in several natural waters (seawater, groundwater, river, pond, and brackish waters) exposed to BAC and beet sugar for about 60 days. They observed that DIC concentrations and $\delta^{13}C$ values in freshwater samples remained unaltered throughout the preservation period. In contrast, salty water samples exhibited DIC changes exceeding the analytical error beyond 15 days. Takahashi and Minami (2022) performed a similar assessment of $^{14}C$ and DIC concentrations in seawater and groundwater. They observed constant $^{14}C$ and DIC concentrations after 30 days of preservation in groundwater samples, while seawater samples experienced increases in both $^{14}C$ and DIC concentrations over time. These studies suggest a common trend: seawater samples treated with BAC remain unaltered for a few days but begin to show changes after one or two weeks. Gloël et al. (2015) examined the impact of BAC addition on the $Ar/O_2$ ratio in dissolved gases in seawater. They reported that seawater samples treated with BAC initially showed values identical to those preserved with $HgCl_2$ for the first 3 or 4 days, but changes emerged after 8–17 days. García et al. (2001) found that 50% of the primary biodegradation of BACs, including benzyl dimethyl tetradecyl ammonium chloride (BAC-$C_{14}$) and benzyl dimethyl hexadecyl ammonium chloride (BAC-$C_{16}$), was completed by marine bacterial populations in 8 to >15 days, respectively. Although they did not identify the specific microorganisms responsible for this degradation, their findings align with the observation that BAC's effectiveness in seawater does not persist long term.

Gloël et al. (2015) noted that the factor diminishes the effectiveness of BAC in seawater over time is likely spores, which are resistant to heat and sterilization. They are highly durable cells that form and lie dormant when bacterial growth conditions deteriorate. Then, as conditions improve because the effectiveness of BAC diminishes, possibly due to interaction with components in the seawater, they can resume growth. The process responsible for reducing BAC's efficacy is unclear, but previous studies mentioned above have suggested that it likely occurs 1–2 weeks after BAC addition to seawater samples. A key factor may be the presence of bacterial spores, but as spores exist universally in both seawater and freshwater (Brown, 2000), their presence alone may cannot fully explain the reduced effectiveness of BAC in seawater compared to freshwater. However, a more practical approach would be to focus on removing the spores present in the water sample. Wilson et al. (2020) demonstrated that filtering water samples to 0.2 μm effectively preserves water for 66 days for $\delta^{13}C$ measurement of DIC. Spores are primarily produced by aerobic *Bacillus* species and anaerobic *Clostridium* species (Brown, 2000), and these rod-shaped bacteria range from approximately 0.2–1 μm in diameter and 1–10 μm in length. As spores are relatively large, they can be easily removed by filtration.

This study aimed to determine an effective method for preventing DIC changes caused by biological activity in seawater samples using BAC addition. We evaluated the effectiveness of BAC treatment alone, filtration alone, and the combined use of BAC and filtration for measuring $\delta^{13}C$ and $^{14}C$ of DIC in both freshwater and seawater samples.

## 2 Materials and procedures

**2.1 Background of filtration**

Filtration can introduce microbubbles, potentially leading to gas dissemination or atmospheric gas exchange. The potential for DIC change during filtration was investigated by comparing the $^{14}$C concentrations of $NaHCO_3$ solutions (1 mmol·L$^{-1}$) before and after filtrations. The $NaHCO_3$ solution used here was prepared by diluting a 1 mol·L$^{-1}$ $NaHCO_3$ reagent solution (Kanto Chemical Co. Inc., Japan), which has a low $^{14}$C concentration of ~0.7 percent Modern Carbon (pMC, Stuiver and Polach,

1977) and a high $\delta^{13}$C value of $-3.8‰$ (Takahashi et al., 2021), with ultrapure water (Milli-Q Direct 8 or Milli-Q Integral 3, Merck Millipore Co., USA). The assessments were carried out using a polyether sulfone (PES) disk filter (25 mm diameter, 0.22 μm pore size, Membrane Solutions Co., Ltd., USA), and a glass fiber (GF) disk filter (25 mm diameter, 1.0 μm pore size, Membrane Solutions Co., Ltd., USA) attached to the syringe (50 mL, disposable syringe, Terumo Corporation, Japan). These samples were designated $NaHCO_3$-unfiltered, $NaHCO_3$-PES, and $NaHCO_3$-GF, respectively. A common technique employed

to impede gas exchange is to introduce the filtered sample water into a sample bottle through a tube, thereby enabling it to overflow. However, in this study, in order to evaluate the maximum impact of filtration, a more drastic technique was employed, whereby the filtrate discharged from the filter was directly poured into a beaker without passing through a tube under atmospheric conditions, thereby exposing the sample to atmospheric $CO_2$. The filtration process was conducted on three occasions, once through a PES filter and twice through a GF filter, using the same $NaHCO_3$ solution, and the resulting three

filtrates were obtained for each filtration treatment. Immediately after the respective filtrations, the filtrates were injected into reaction containers to extract $CO_2$ from water sample. To ascertain whether any DIC was altered during the filtration experiment, one unfiltered $NaHCO_3$ solution each was introduced into the reaction containers before and after the three filtration treatments (Fig. S1).

**2.2 Filtration, BAC addition, and combined treatment**

To compare the efficacy of the various treatments, the following six conditions were evaluated using the natural water samples (SW and GW) mentioned below: (1) no filtration, no BAC addition (Control), (2) BAC addition alone (BAC), (3) filtration through a PTFE filter (PTFE), (4) filtration through a PES filter (PES), (5) PTFE filtration with BAC addition (PTFE+BAC), and (6) PES filtration with BAC addition (PES+BAC). For each assessment of six treatments, the changes of $^{14}$C and $\delta^{13}$C during the preservation were quantified on a single occasion. The SW was collected from the sea surface of the Pacific coast

at the Nagoya Port, located in Nagoya City, Aichi Prefecture, Japan. This site is located near the estuaries of three rivers: the Nikko, Shinkawa, and Shonai rivers, in a tidal flat region. The GW was obtained from a well of 80 m deep in Tsukuba City, Ibaraki Prefecture, Japan. The chemical composition of the SW (Table S1) indicated that the SW was diluted by river water. The salinities of the SW and GW were determined by summing of the chemical composition data to be 204 and below 0.5, respectively (Table S1). While SW can be considered as a brackish water sample, it is treated as a coastal seawater sample in

this study. For the SW, the expected value of DIC concentration is slightly smaller than 2 mmol·L$^{-1}$ and that of $^{14}$C concentration is ~100 pMC. The DIC and $^{14}$C concentrations of groundwater taken from the same well were reported to be 1.69 mmol·L$^{-1}$ and 21.3 ± 0.1 pMC, respectively (Takahashi et al., 2019a; Takahashi and Minami, 2022). At the time of sampling, neither disinfectant treatment nor filtration was applied to any of the natural water samples.

To promote microbial activity and detect even minor changes in DIC that might result from biological processes, beet sugar

powder was added in the sample water at a concentration of 2 g·L$^{-1}$ before the preservation of the sample. Given the high $^{14}$C concentration of 103.3 ± 0.7 pMC in beet sugar (Takahashi and Minami, 2022), which is approximately equivalent to that of SW, it is conceivable that any $^{14}$C changes resulting from the microbial decomposition of beet sugar to DIC might be undetectable in SW samples. To address this, $NaHCO_3$ solution (1 mol·L$^{-1}$ solution, Kanto Chemical Co. Inc., Japan), which has a low $^{14}$C concentration of ~0.7 pMC and a high $\delta^{13}$C value of $-3.8‰$ (Takahashi et al., 2021), was added to the samples.

The addition of NaHCO$_3$ also helped to clarify the $\delta^{13}$C changes in DIC due to beet sugar decomposition, as beet sugar exhibits a low $\delta^{13}$C value of −26.2‰ (Takahashi and Minami, 2022). Unless otherwise stated, beet sugar and NaHCO$_3$ solution were added to all water samples.

The treatments were conducted in the following sequence: NaHCO$_3$ reagent solution (1 mol·L$^{-1}$) was added to the sample waters (~4 L) at a rate of 2.5 mL·L$^{-1}$ of SW and 2 mL·L$^{-1}$ of GW, effectively doubling the DIC concentration in both water

types. This is (1) Control sample. For assessment (2) BAC, BAC (10% solution, FUJIFILM Wako Pure Chemical Co., Japan) was added to an aliquot of the sample at a concentration of 0.01%. Then, the sample waters not used in assessment (2) were filtered using a PTFE or PES disk filter (25 mm diameter, 0.22 μm pore size, Membrane Solutions Co., Ltd.) attached to a syringe (50 mL, disposable syringe) to a beaker. The first 20–30 mL of filtrate was not used for the assessment to pre-wash the filters. These samples were directly utilized for assessments (3) PTFE and (4) PES. For assessments (5) PTFE+BAC and

(6) PES+BAC, BAC was added to the retained filtrate at a concentration of 0.01%. All the treated samples were homogenized in a beaker using a magnetic stirrer. They were then injected into one reaction container for CO$_2$ extraction to obtain initial $^{14}$C concentration and $\delta^{13}$C value, also injected into three preservation bottles (125 mL glass vials) sealed with butyl rubber septa coated with Teflon, and aluminium caps, which had been filled with beet sugar powder and evacuated. The water injection into the reaction container and preservation bottles was carried out through a needle attached to a syringe immediately

after sample treatments. The preservation periods were 14, 28, and 285 days for SW, and 14, 28, and 126 days for GW. At the end of each preservation period, the bottles were opened one by one, and the CO$_2$ was extracted (Fig. S2). Before experiments, the vials and butyl rubber septa were sterilized by heating at 450ºC for 6 h and by autoclaving, respectively. The BAC used in this study primarily consisted of benzyl dimethyl dodecyl ammonium chloride (BAC-C$_{12}$) and benzyl dimethyl tetradecyl ammonium chloride (BAC-C$_{14}$).

Since the filters for treatments (3) – (6) were not sterilized, additional assessments were conducted using sterilized PES disk filters (25 mm diameter, 0.2 μm and 0.45 μm pore sizes, GVS Japan). GW samples, with or without filtration, were preserved in 34 mL glass vials for 6, 14, and 28 days. These treatments were labelled GW-Control2, GW-PES2 (0.2 μm), and GW-PES2 (0.45 μm), respectively (Fig. S3). A total of 12 bottles were prepared for each treatment, with three bottles used for the initial value and each of the three preservation periods to measure $\delta^{13}$C values by GC-IRMS. As these samples were not analyzed for

$^{14}$C, NaHCO$_3$ solution to lower the initial $^{14}$C concentration was not added. Other procedures were same as treatments (3) and (4).

## 2.3 $^{14}$C concentration and $\delta^{13}$C measurements

CO$_2$ extraction from water samples for the measurement of $^{14}$C concentration and $\delta^{13}$C values was performed using the ReCEIT (repeated cycles of extraction, introduction, and trapping) method (Takahashi et al., 2021), which is a simple and carrier gas-

free method that can handle a variety of water samples with a wide range of DIC concentrations (0.4–100 mmol/L, in the case of 1 mmol of carbon) and produces high CO$_2$ yields. The procedure is composed of repeating the cycles of CO$_2$ extraction from water into the headspace of the reaction container, expansion of the extracted gas into the vacuum line, and cryogenic trapping of CO$_2$. This method, which extracts CO$_2$ without bubbling, is particularly well-suited for BAC-added samples, which tend to foam. An approximate DIC concentration was calculated from the volume of water treated and the CO$_2$ extracted. The

CO$_2$ gas was reduced to graphite (Kitagawa et al., 1993) for analysis by accelerator mass spectrometry (AMS), following the removal of sulfur oxide gas using the Sulfix reagent (8–20 mesh, Kishida Chemical Co., Ltd., Japan) as necessary. The $^{14}$C concentrations were measured using a 3 MV AMS (Model 4130-AMS, HVEE, Netherlands) at the Institute for Space–Earth Environmental Research, Nagoya University, Japan (Nakamura et al., 2000) and a 1 MV AMS (4110Bo-AMS-3, HVEE, Netherlands) at the Korea Institute of Geoscience and Mineral Resources (KIGAM), Korea (Hong et al., 2010). Corrections

for isotopic fractionation (Stuiver and Polach, 1977) were performed using the $^{13}$C/$^{12}$C ratio measured by AMS. The standard deviations for $^{14}$C measurements were 0.02–0.04 pMC for waters with concentrations below 1 pMC, and less than 0.8% of $^{14}$C

concentration for waters above 10 pMC when measured at the Nagoya University. The precision of the quantitative analysis of carbon was better than 3%, and the background was below $2 \times 10^{-15}$ at KIGAM. The error of $^{14}$C measurement was represented in 1σ based on a standard calculation method in $^{14}$C analysis (Scott et al., 2007).

The δ$^{13}$C values of $CO_2$ gas extracted by the ReCEIT procedure were determined via isotope-ratio mass spectrometry (IRMS) with a dual inlet system (Delta-V Advantage, Thermo Fisher Scientific, Inc., USA) at the Geological Survey of Japan. The standard deviation of multiple δ$^{13}$C measurements by IRMS is less than 0.01‰, and for individual measurements, the error is represented as 1σ calculated from the variations in the dual inlet measurement. Some δ$^{13}$C measurements were performed using a continuous-flow IRMS coupled to a gas chromatography system (GC-IRMS; Delta-V Advantage with Gas Bench II,

Thermo Fisher Scientific, Inc., USA) at the Geological Survey of Japan (Takahashi et al., 2019b). The standard deviation of multiple measurements of water samples by GC-IRMS is 0.04‰ (1σ). $CO_2$ gas was extracted from water by addition of phosphoric acid in a septum-sealed Exetainers® vial (12 mL, Labco Ltd., UK). The δ$^{13}$C value of each sample is represented as the averaged value over the three vials with the standard deviation (1σ).

## 3 Results and discussion

### 3.1 Background on the $^{14}$C concentration and δ$^{13}$C values in the filtration treatment process

The $^{14}$C concentrations and δ$^{13}$C values of two $NaHCO_3$-unfiltered samples, both before and after filtration assessments, were identical, indicating that the DIC of the $NaHCO_3$ solution itself remained unchanged during the assessment experiment (Fig. 1, Table S2). The $^{14}$C concentrations of $NaHCO_3$-unfiltered, $NaHCO_3$-PES, and $NaHCO_3$-GF were consistent within the error range. Although each value was obtained from a single experiment, the five analytical results showed high consistency,

indicating no detectable change in $^{14}$C concentration due to filtration. This suggests that any increase in $^{14}$C due to $CO_2$ contamination during filtration was not a significant concern. In contrast, the δ$^{13}$C values showed a very slight decrease for $NaHCO_3$-PES and $NaHCO_3$-GF (Fig. 1, Table S2). This slight change in δ$^{13}$C is assumed to be caused by atmospheric $CO_2$ contamination or $CO_2$ degassing from the $NaHCO_3$ solution during filtration. If atmospheric $CO_2$ contamination caused the δ$^{13}$C shift in the $NaHCO_3$ solution, the $^{14}$C concentration would vary according to the amount of atmospheric $CO_2$

contamination. Assuming atmospheric $CO_2$ has a δ$^{13}$C value of −10‰ and a $^{14}$C concentration of 100 pMC, the $^{14}$C concentrations of $NaHCO_3$-PES and two $NaHCO_3$-GF could be calculated as $3.6 \pm 0.2$ pMC, $4.5 \pm 0.2$ pMC, and $2.4 \pm 0.2$ pMC, respectively. These calculated values do not align with the measured $^{14}$C concentrations, suggesting that atmospheric $CO_2$ contamination is unlikely.

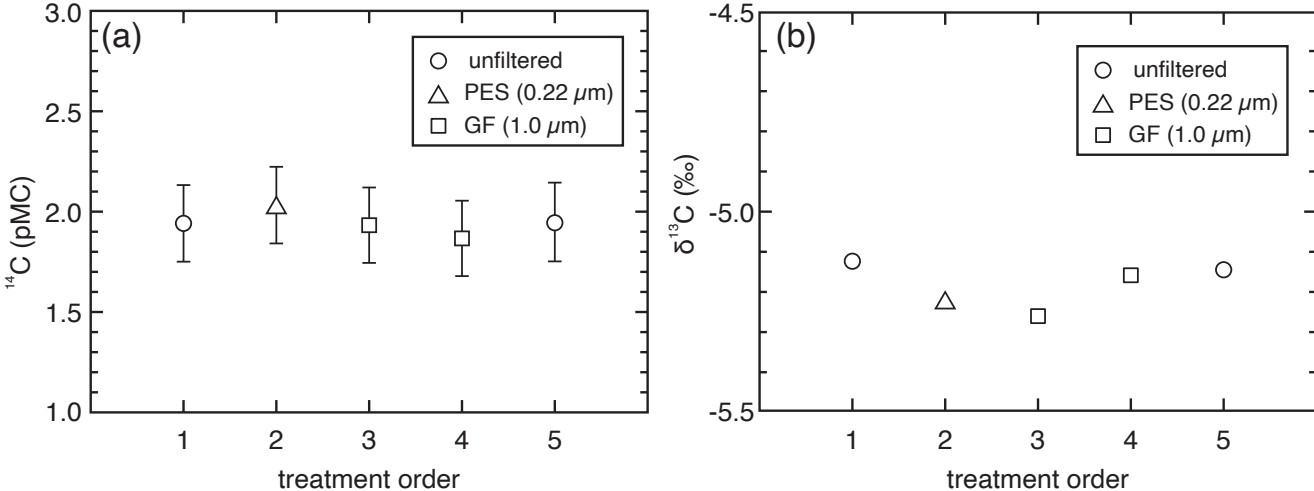

Figure 1: Comparisons of $^{14}C$ (a) and $\delta^{13}C$ (b) among the unfiltered and filtered solutions of 1 mmol·L$^{-1}$ of NaHCO$_3$. PES: filterd by PES disk filter (25 mm in diameter, 0.22 μm in pore size), GF: filtered by GF disk filter (25 mm in diameter, 1.0 μm in pore size). Each value of $^{14}C$ and $\delta^{13}C$ represents a single treatment occasion. The bars of $^{14}C$ concentration represent the measurement error of the AMS analysis. The $\delta^{13}C$ error is smaller than the size of the plotting symbols.

The $\delta^{13}C$ of DIC would change due to isotope fractionation associated with degassing. When DIC and gaseous $CO_2$ are in isotopic equilibrium, the $\delta^{13}C$ of DIC is typically higher than that of gaseous $CO_2$ (Zhang et al., 1995). As carbon with a lower $\delta^{13}C$ value is removed as $CO_2$ during degassing, the $\delta^{13}C$ of the remaining DIC in the solution would gradually increase. However, the measured $\delta^{13}C$ showed the opposite trend, indicating that the change in $\delta^{13}C$ is not due to $CO_2$ degassing from the NaHCO$_3$ solution. Thus, the two hypotheses—that $\delta^{13}C$ changes were caused by atmospheric $CO_2$ contamination or by degassing—were rejected, and the observed $\delta^{13}C$ change may be attributed to an unidentified artifact factor other than filtration. Carbon contamination during sample treatments could significantly influence $^{14}C$ analysis, but the impact of isotopic fractionation can be eliminated by corrective calculations. Since the primary objective of this study is $^{14}C$ analysis, the effect of filtration is expected to be minimal or negligible. However, if $\delta^{13}C$ analysis were conducted, careful scrutiny and verification would be necessary. The experimental procedure in this study was designed to evaluate the maximum impact of filtration, and it is anticipated that the impact can be minimized by adopting experimental procedures that minimize filtration-related effects. As the filtration in this assessment was performed under atmospheric conditions with $CO_2$ exposure, it was likely to cause carbon contamination. However, the identical $^{14}C$ concentrations (Fig. 1) suggest that a $^{14}C$ increase due to $CO_2$ contamination during filtration should not be considered a concern. Nonetheless, depending on the filter material or pore size, the water sample may not pass through unless the syringe is pressed forcefully, which can lead to contamination by the atmospheric $CO_2$ inside the syringe. When filtration was performed with a 1 mmol·L$^{-1}$ NaHCO$_3$ solution and an equal volume of air inside the syringe using a PES filter (0.22 μm), the $^{14}C$ concentration of the NaHCO$_3$ solution was measured to increase by 0.7 pMC, rising to 4.6 pMC from an initial 3.9 pMC in our assessment. This $^{14}C$ increase is quantitatively reasonable, assuming a $CO_2$ concentration of 400 ppm and that the $CO_2$ inside the syringe fully dissolved into the NaHCO$_3$ solution. It is important to remove air bubbles in the syringe at the filtration.

### 3.2 $^{14}C$ concentration and $\delta^{13}C$ changes in natural water samples

The initial values of DIC concentrations, $^{14}C$ concentrations, and $\delta^{13}C$ values for SW mixed with NaHCO$_3$ solution were 3.60–3.65 mmol·L$^{-1}$, 41.2–42.2 pMC, and −4.05 to −3.72‰, respectively (Table S3). For GW, these values were 4.48–4.93 mmol·L$^{-1}$, 10.2–10.9 pMC, and −6.00 to −5.98‰ when mixed with NaHCO$_3$ solution, and 1.85–1.87 mmol·L$^{-1}$ and −7.74 to −7.69‰ when not mixed with NaHCO$_3$ solution (Table S1), respectively. After mixing with NaHCO$_3$, the $^{14}C$ concentrations in both SW and GW were approximately half or slightly less than half of their original concentrations.

The largest changes in $^{14}$C and $\delta^{13}$C during the preservation period were observed in the Control samples, with progressively smaller changes occurring in the order of filtration-only samples, BAC-only samples, and those treated with both filtration and BAC. The $^{14}$C concentrations increased as the preservation period lengthened for SW-Control, GW-Control, SW-PES, GW-PES, SW-PTFE, GW-PTFE, and SW-BAC (Fig. 2). It is reasonable to assume that these large changes in $^{14}$C concentration and $\delta^{13}$C were caused by the DIC derived from beet sugar, given that beet sugar is more easily degraded than BAC or other organic materials suspended in water. Given that DIC change during the preservation was enhanced by the incorporation of sugar, it is imperative to ascertain the impact of sugar addition. Takahashi and Minami (2022) defined the boost effect as an index of how many times the DIC change in the sugar-added sample is greater than the DIC concentration change in the no-sugar sample during the preservation. It is anticipated that the boost effect will be more pronounced in instances where there is a paucity of organic matter and a greater prevalence of microorganisms in the water sample. The SW in this study was sampled at a tidal flat location along the Pacific coast, near the estuaries of major rivers. It can be reasonably assumed that water discharged from tidal flats will have higher concentrations of organic carbon, nutrients, and microbes than typical seawater (Sakamaki et al., 2006; Hu et al., 2016). Accordingly, the boost effect of SW in this study may be identical to or slightly smaller than $3.0 \pm 1.4$, as reported by Takahashi and Minami (2022) for the seawater sample sampled at Kashima Port on the Pacific coast. This seawater was not mixed with river waters and was not sampled from the tidal flat. The boost effect of groundwater sampled from the same well as the GW was reported to be $5.3 \pm 1.8$. As the exact boost effect is not measured in the present study, the increase in DIC due to sugar addition was not corrected through calculation. It is important to note that the described DIC change includes an increase of probably two or three times for SW and five times for GW caused by the addition of sugar.

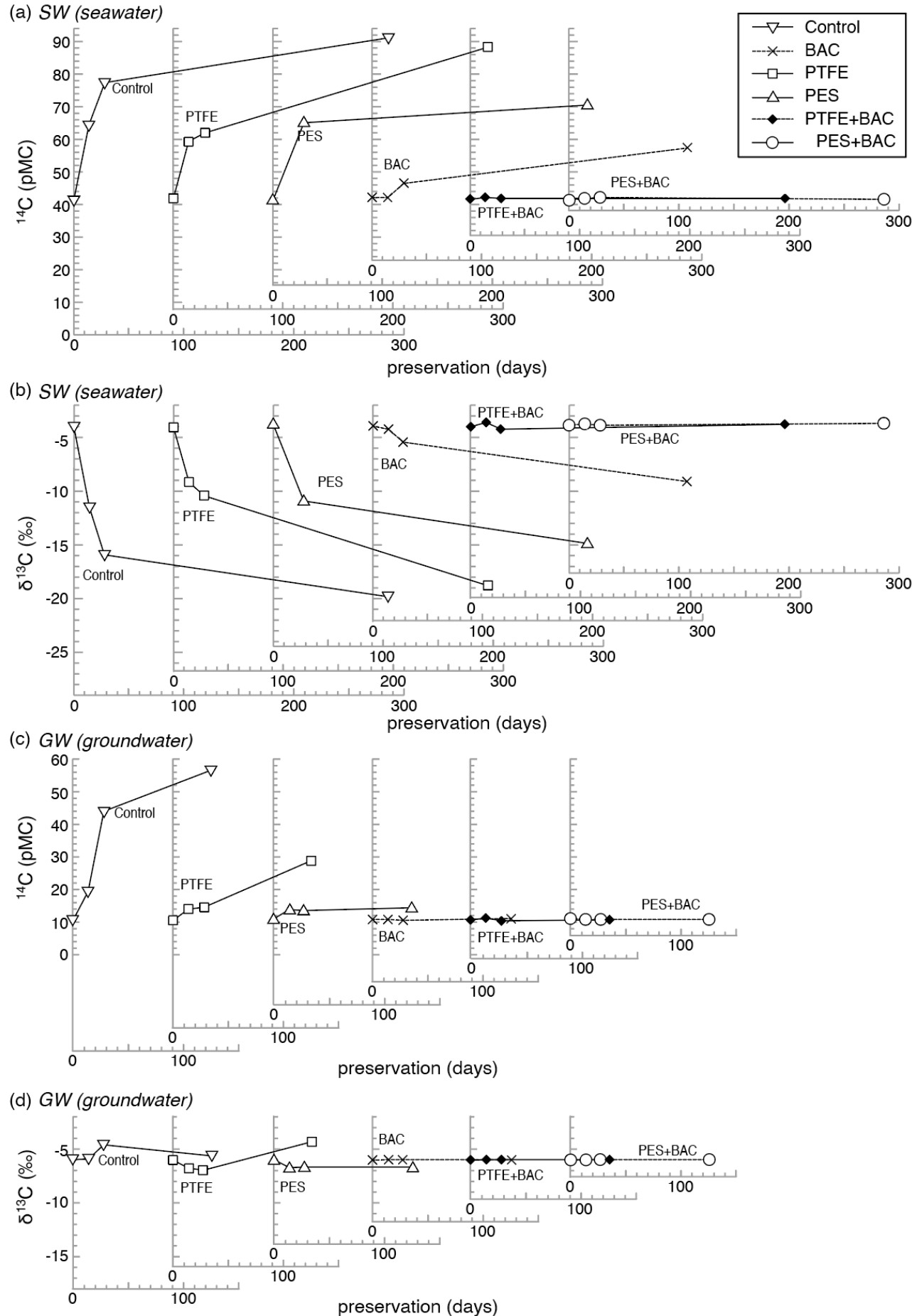

Figure 2: Changes in $^{14}$C and $\delta^{13}$C during the preservation of SW and GW mixed with NaHCO$_3$ solution and beet sugar. Changes in DIC were augmented by the addition of beet sugar. (a) $^{14}$C of SW, (b) $\delta^{13}$C of SW, (c) $^{14}$C of GW, (d) $\delta^{13}$C of GW. The time series of DIC change for the respective treatments were derived from each experiment conducted on a single sample with varying preservation periods. The errors of $^{14}$C and $\delta^{13}$C are equivalent to or smaller than the size of the plotting symbols.

## 3.3 Comparison of treatments

Filtration, BAC addition, and the combined treatment indicated more effective than the Control samples in preserving DIC in water samples (Fig. 2). However, in some cases, changes could not be completely reduced, depending on the type of treatment or the water used. In samples treated with BAC alone, the results were consistent with previous studies (Takahashi et al., 2019b; Takahashi and Minami, 2022), where GW showed suppression of DIC changes, but SW showed changes of 15.1 pMC in $^{14}$C concentration and $-5.2‰$ in $\delta^{13}$C. This confirms that BAC alone is not suitable for seawater samples and indicates that the seawater sample utilized in this investigation contains constituents, probably unique microorganisms, whose biological activity cannot be entirely suppressed by BAC addition, as reported in previous studies. Accordingly, SW represents an appropriate sample for the purpose of investigating potential methods for addressing the issue of BAC impairment in seawater. The $^{14}$C concentrations and $\delta^{13}$C values were observed to be relatively constant in samples treated with both filtration and BAC: SW-PTFE+BAC, SW-PES+BAC, GW-BAC, GW-PTFE+BAC, and GW-PES+BAC. The $^{14}$C and $\delta^{13}$C changes were minimal for SW-PTFE+BAC, SW-PES+BAC, GW-BAC, GW-PTFE+BAC, and GW-PES+BAC (Table S3). Since only a single time series of data is available for each treatment, the actual values for DIC changes remain uncertain. However, as the comparison is based on time series data, it can be posited that treatments exhibiting minimal change are highly effective.

The $^{14}$C concentration and $\delta^{13}$C value of filtered waters without BAC addition showed significant changes, although they were smaller than those in unfiltered samples. The $^{14}$C changes of SW-Control, SW-PTFE, and SW-PES were 50.0 pMC, 46.4 pMC, and 29.2 pMC, respectively, while those of GW-Control, GW-PTFE, and GW-PES were 46.1 pMC, 18.3 pMC, and 3.6 pMC, respectively. DIC changes were smaller with PES filtration than PTFE for both SW and GW, despite using the same pore-size filter. This may be related to the fact that PTFE has higher resistance than PES, requiring more force during filtration.

The changes in $\delta^{13}$C of the GW-PES2 as the sample filtered through a sterile filter were consistent with those of GW-Control2 as the unfiltered sample, except for a slight change of GW-PES2 (0.2 μm) after 6 days (Table 1). This suggests that DIC changes of the filtered samples shown in Fig. 2 were not caused by the microorganisms derived from the filter. While using a 0.2 μm filter reduces the number of microorganisms compared with a 0.45-μm filter, once some slip through, the difference between filters may disappear as the microorganisms proliferate. Wilson et al. (2020) reported that filtration alone is sufficient to prevent DIC changes due to its biological activity. However, our results showed that this method was insufficient, as DIC changes could not be ignored for SW-PTFE, SW-PES, and GW-PES2 (0.2 μm) with preservation periods longer than 14 days, although this study confirmed that filtration reduces DIC changes in water samples during preservation (Fig. 2, Table 1). This was especially the case for GW-PES2 (0.45 μm) samples, where sugar addition increased biological activity. Sugar addition may have artificially triggered microbial growth, resulting in DIC changes that would not have occurred otherwise. Without microbial growth triggers, filtration may be more effective, but DIC changes were smaller with SW-BAC compared to SW-PTFE or SW-PES. Therefore, BAC addition is more effective than filtration alone in reducing DIC changes, although it has the disadvantage of not being able to use the sample for other analyses.

**Table 1: Initial values of DIC concentrations and mean values of $\delta^{13}$C with the standard deviation (1σ, N=3) for GW-Control2, GW-PES2 (0.2 μm), and GW-PES2 (0.45 μm), and the changes in $\delta^{13}$C values with propagation error during the preservation.**

| sample | DIC initial (mmol·L$^{-1}$) | $\delta^{13}$C initial (‰) | $\delta^{13}$C change from initial (‰) | | |
| --- | --- | --- | --- | --- | --- |
| | | | *6 days* | *14 days* | *28 days* |
| GW-Control2 | 1.87 | $-7.74 \pm 0.06$ | $-1.88 \pm 0.13$ | $-2.15 \pm 0.73$ | $-2.07 \pm 0.60$ |
| GW-PES2 (0.2 μm) | 1.85 | $-7.70 \pm 0.07$ | $-0.06 \pm 0.08$ | $-2.05 \pm 0.08$ | $-1.99 \pm 0.24$ |
| GW-PES2 (0.45 μm) | 1.85 | $-7.69 \pm 0.10$ | $-2.28 \pm 0.47$ | $-2.09 \pm 0.12$ | $-2.20 \pm 0.41$ |

## 3.4 Combined treatment of BAC addition and filtration

The combined treatments, PTFE+BAC and PES+BAC, showed consistent [14]C concentrations within the analytical error during preservation for both SW and GW. Though slight $\delta^{13}C$ changes were observed in SW, but this $\delta^{13}C$ change seems to be negligible given its small magnitude and uncertainty which only became detectable through sugar-induced microbial activity magnification.

One possible explanation for the minimal DIC changes in the combined treatment may be due to effectiveness of BAC was enhanced by reducing the number of microorganisms through filtration. This explains why no DIC changes were observed in SW-PTFE+BAC and SW-PES+BAC during the preservation period. In contrast, the DIC change observed in SW-BAC may be caused by BAC being insufficient against the number of microorganisms present, however, the reason why the change was observed during the only second half of the preservation period cannot be explained. If microorganisms are killed by BAC, reactivation should not occur in the second half of the preservation period. As mentioned in the Introduction, it has been suggested that the lower effectiveness of BAC in seawater may be due to spores (Gloël et al., 2015), that cannot be effectively inactivated by BAC. They are not significantly different in size from rod-shaped bacteria, the main spore-forming microorganisms (Brown, 2000), and range in size from 0.6–4 µm (Reponen et al., 2001). These larger microorganisms are likely to be removed by filtration. Our assessment indicated that filtration alone might allow some microorganisms to pass through, but BAC can inactivate the vegetative cells of small microorganisms. Filtration removes larger spores that may cause DIC changes in seawater. The role of spores has not been fully verified, but this could explain why DIC in SW-BAC remained unchanged until 14 days and changed after 28 days.

Certain bacteria can degrade QACs (García et al., 2001; Patrauchan and Oriel, 2003; Zhang et al., 2011; Oh et al., 2013). If the water sample contained microorganisms capable of degrading BAC, biological activity would not be inhibited, leading to DIC changes. While BAC may kill most microorganisms, BAC-tolerant microorganisms could survive and recover, causing detectable DIC changes. This could explain the DIC change observed in SW-BAC. If BAC-tolerant microorganisms were removed by filtration, it would align with the lack of DIC changes in SW-PTFE+BAC and SW-PES+BAC. Microorganisms reported to adapt to BAC and cause biodegradation include *Pseudomonas* spp., *Aeromonas hydrophila*, *Salmonella enterica*, and *Klebsiella oxytoca* (Ferreira et al., 2011; Khan et al., 2015; Cui et al., 2023). These species are commonly found in aquatic environments. These microorganisms are not originally tolerant to BAC but gradually adapt over long periods, such as several tens of days of exposure (Oh et al., 2013; Yang et al., 2023). Preservation in sealed vials without aeration for DIC analysis is unlikely to permit the adaptation of microorganisms to BAC due to insufficient exposure time. Thus, for BAC biodegradation to occur, microorganisms must be initially tolerant of BAC in the water sample. QACs in water are removed by microbial communities tolerant to them, often found in sewage treatment plants (Zhang et al., 2015; Deleo et al., 2020). Previous biodegradation studies have isolated microbial communities from enriched cultures grown on BAC-based media or activated sludge (Chacón et al., 2023). Therefore, it cannot be ruled out that microorganisms tolerant to BAC might be present in estuary and coastal waters where sewage effluents mix. However, degradation of BAC, a refractory organic compound, only begins after readily decomposable organic matter is fully consumed (Zhang et al., 2011). If BAC-tolerant microorganisms were present, beet sugar would be consumed first, and the [14]C and $\delta^{13}C$ of DIC in preserved water samples would reflect beet sugar more closely than BAC. Even in this case, the lack of DIC changes in the SW-PTFE+BAC and SW-PES+BAC samples indicates that filtration is effective at removing such microorganisms. When combining filtration and BAC addition, avoiding contamination with ambient carbon (atmospheric $CO_2$) during filtration is essential. A blank check with $NaHCO_3$ demonstrated that the [14]C background remained unchanged, confirming that filtration represents a viable process provided that the necessary precautions are properly followed. As a consequence of ongoing technological advancements, the quantity of carbon required for [14]C measurement by AMS is progressively diminishing (Minami et al., 2013; Ruff et al., 2010). The reduction in sample size facilitates filtration and minimizes background contamination, which represents a favourable development for this combined procedure.

# 4 Conclusions

This study assessed several treatments aimed at suppressing changes in DIC during sample preservation as an alternative to HgCl₂ addition. We found that the combination of filtration and BAC addition effectively inhibited DIC changes due to microbial activity during preservation. In water samples treated with this method, DIC changes were minimal, even when sugar was added to significantly enhance microbial activity. In practical analyses, such a boost from adding sugar would not occur, leading to the conclusion that the combined method of BAC addition and filtration is an effective procedure for

inhibiting DIC changes caused by biological activity. The slight changes in DIC observed in BAC-supplemented seawater samples may be attributed to microorganisms that BAC could not inactivate. They are presumed to be spores or BAC-tolerant microorganisms. The size of spores and spore-forming microorganisms varies among species, but they are generally large enough to be removed through filtration. Many microorganisms that adapt to BAC are also relatively large and can be removed by filtration.

DIC changes in seawater samples can be suppressed through a two-step process: first, filtration to remove spores or microorganisms tolerant to BAC (if present), followed by BAC treatment to inactivate microorganisms that passed through filtration. It should be noted that smaller microorganisms may still pass through the filtration system. In some cases, the partial removal of microorganisms through filtration may not fully suppress the DIC changes, leading to microorganism recovery within a few days, resulting in DIC changes similar to those in the unfiltered samples. Therefore, careful consideration of the

preservation period is necessary when using filtration alone to suppress DIC changes.

  We recommend a combined treatment of filtration and BAC addition to suppress DIC changes during sample preservation (Fig. 3), as it offers a safer alternative to HgCl₂. In this study, a 0.22 μm pore size filter was used to validate earlier findings. However, it is likely that a filter with a coarser pore size could also remove spores, given their size. Verifying the optimal pore size is an important next step. As observed in this study, using a very fine pore-size filter can slow sample flow, increase

resistance, impair operability, and elevate blanks. Future research should include blank verifications of the combined technique and further verification of whether the slight DIC change observed here can be detected in other water samples. It should also be noted that BAC in water may be removed primarily by adsorption onto sludge rather than by biodegradation (Zhang et al., 2015). Our preliminary result showed that the bactericidal efficiency of BAC was likely diminished in water sample containing muddy sediment. While filtration can remove sludge or sediment, caution is needed when applying the combined treatment to

water samples containing large amounts of suspended material. In such cases, increasing the amount of BAC may be necessary for effective treatment. The assessment of this combined procedure was conducted on a limited number of natural water samples, and further investigation into the optimal filter pore size and verification using other natural water samples is necessary. However, this procedure offers a practical and environmentally friendly alternative to conventional mercury-disinfected methods for water sample preservation in aquatic environments.

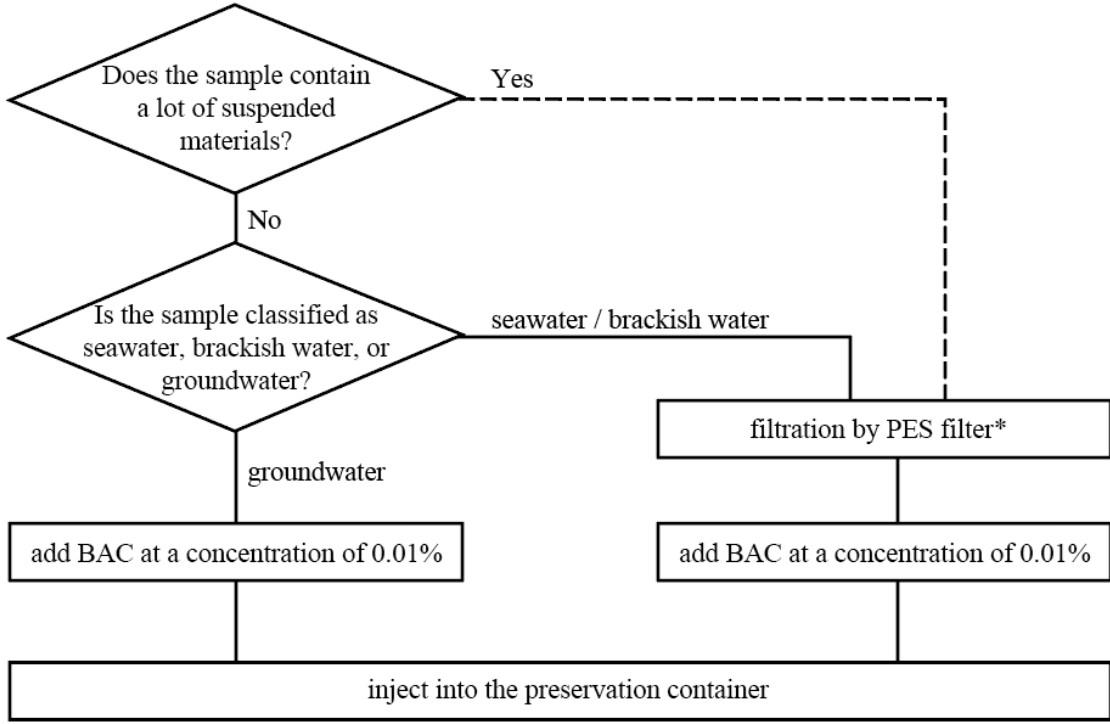

**Figure 3: Summary flowchart for the preservation of water samples for radiocarbon measurement in DIC. Dotted line: Although not directly verified in this study, filtration is recommended based on the reporting of Zhang et al. (2015), which suggests that BAC can be removed by adsorption to particles in water. \*: Filters with a pore size of 0.22 μm were evaluated in this study; while PTFE filters were effective, they showed significantly higher resistance to filtration compared to PES filters. As a result, PES filters are expected to induce fewer changes in DIC.**

**Data availability.** The data utilized in this study are presented in Tables in the manuscript and the supplementary material.

**Author contribution.** HAT participated in the design and discussion of the study, as well as in the $\delta^{13}C$ measurements. MM contributed to the discussion and to the $^{14}C$ measurements.

**Competing interests.** The author declares that he has no conflict of interest.

**Acknowledgements.** The authors thank Professor Hiroyuki Kitagawa of Nagoya University and Dr. Wan Hong of KIGAM for their help with the AMS measurements. We are also grateful to Ms. Hiroko Handa of the Geological Survey of Japan, AIST (affiliated at the time of the experiment) and Mr. Koh Kakiuchida of Nagoya University for their experimental support of $CO_2$ extraction, and Mr. Akihiko Inamura of the Geological Survey of Japan, AIST for his help with the ion chromatography measurements. The authors also sincerely thank the anonymous reviewers for their time and thoughtful feedback on our manuscript. Part of this study was carried out by the joint research program of the Institute for Space–Earth Environmental Research, Nagoya University. This study was supported by JSPS KAKENHI, Grant Number 23K03500.

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
