# Peer review of "Combining benzalkonium chloride addition with filtration to inhibit dissolved inorganic carbon alteration during the preservation of water sample in radiocarbon analysis"

_EGUsphere, 2024_

## Author Comment (AC2)

[Figure]

**Figure S1: Flowchart of experimental treatments to evaluate filtration background. The treatment was carried out in the order of the numbers in parentheses.**

[Figure]

**Figure S2: Flowchart of experimental treatments to evaluate six treatment comparison.**

[Figure]

**Figure S3: Flowchart of experimental treatments to evaluate filtration by sterilized filter.**

---

## Author Response (AR1)

We would like to thank the anonymous reviewers for their time and thoughtful feedback on our manuscript. We have revised the manuscript according to their comments. Our detailed responses to the reviewers' comments are provided below. We believe that we have addressed all of the comments. In the table below, additions to the text are highlighted in red and deletions are shown in blue with strikethrough. The line numbers correspond to those in the revised manuscript.

| Comment from reviewer | Answer |
|---|---|
| <RC1> The experimental design needs to be made explicitly clear. The manuscript must state how many independent replicates were used for each treatment in order for the reader to better understand what the results were. At present, it appears n = 1 or 2 for some or all experiments which is too low to warrant publication. | The description of the experimental design has been revised to include the number of samples processed or measured. The measured values, excluding GW-Control2, GW-PES2 (0.2 µm), and GW-PES2 (0.45 µm), were obtained from a single measurement of an individual treatment or a preservation bottle. The three excluded samples are presented as the mean and standard deviation of the analytical values from the three preservation bottles of each. The revised description includes both the number of samples, and their errors.

 In response to the comment suggesting that the number of analyses is insufficient, we present the following counterargument:
 Background assessment: While not previously delineated in the initial manuscript, the filtration background assessment was conducted using a single NaHCO$_3$ solution, with the order of unfiltered, PES, GF, GF, and unfiltered samples. The $^{14}$C concentrations and $\delta^{13}$C values of the NaHCO$_3$ solution before and after the filtration experiment were identical, indicating that the DIC of the NaHCO$_3$ solution itself was not affected during the evaluation experiment. A comparison of filtered and unfiltered samples showed a high degree of consistency in analytical values, indicating that the filtration did not change the $^{14}$C concentration. While the individual data points were obtained from a single experiment, the results of the assessment should be evaluated as being derived from the results of a series of experiments. Therefore, we believe it is reasonable to conclude that the results are not merely due to chance but reliably indicate the absence of an influence of filtration. The revised manuscript includes the sample processing order for both unfiltered and filtered samples and a discussion of these results.
 Comparison of treatments: The results were obtained as a time-series, although the individual data points were obtained from a single experiment. While the detailed values pertaining to isotopic changes may not be entirely accurate, it is possible to discern the presence or absence of alterations and trends. If the objective is to ascertain an exact amount of change, repeated measurements are necessary. However, if the objective is to determine whether change has occurred or to identify its direction of change, we believe that the results of this study are sufficient for this purpose. |
| <RC2> The results of these different treatment are as somewhat expected, that filtration with small pore filter and bacteria inhibition help to preserve DIC samples, but with limited success (i.e., relatively short storage time). However, the experiment section needs much articulation based on what is presented in the manuscript. A flow chart that includes many of the treatment details may be helpful for readers to comprehend the procedure in the experiment. For example, please include the volume of the initial sample vessel, how samples were distributed into these different treatments, and number of replicate samples/analysis (N) for each treatment. | We initially anticipated that filtration alone would not have enough effect to inhibit DIC changes during water preservation. However, a recent publication asserted that treatment of only the filtration was effective, prompting further investigation. The findings of the present study, while consistent with the initial predictions, also revealed the necessity of employing both filtration and BAC in the context of seawater samples.
 The Materials and procedures section has been revised to clearly explain the experiments, especially the number of samples. The flowchart has been added as the supplement materials (Figs. S1 – S3). |
| Revision for corresponding to above | |
| Lines 80–98 | 2.1 Background of filtration
 Filtration can introduce microbubbles, potentially leading to gas dissemination or atmospheric gas exchange.  $^{14}$C analysis. The potential for DIC change  during filtration was investigated by comparing the $^{14}$C concentrations of NaHCO$_3$ solutions (1 mmol·L$^{-1}$) before and after filtrations. The NaHCO$_3$ solution used here was prepared by diluting a 1 mol·L$^{-1}$ NaHCO$_3$ reagent solution (Kanto Chemical Co. Inc., Japan), which has a low $^{14}$C concentration of ~0.7 percent Modern Carbon (pMC, Stuiver and Polach, 1977) and a high $\delta^{13}$C value of −3.8‰ (Takahashi et al., 2021), with ultrapure water (Milli-Q Direct 8 or Milli-Q Integral 3, Merck Millipore Co., USA). The assessments were carried out using a polyether sulfone (PES) disk filter (25 mm diameter, 0.22 µm pore size, Membrane Solutions Co., Ltd., USA), and a glass fiber (GF) disk filter (25 mm diameter, 1.0 µm pore size, Membrane Solutions Co., Ltd., USA) attached to the syringe (50 mL, disposable syringe, Terumo Corporation, Japan). These samples were designated NaHCO$_3$-unfiltered, NaHCO$_3$-PES, and NaHCO$_3$-GF, respectively. A common technique employed to impede gas exchange is to introduce the filtered sample water into a sample bottle through a tube, thereby enabling it to overflow. However, in this study, in order to evaluate the maximum impact of filtration, a more drastic technique was employed, whereby the filtrate discharged from the filter was directly poured into a beaker without passing through a tube.  under atmospheric conditions, thereby exposing the sample to atmospheric CO$_2$. The filtration process |

[revised manuscript text omitted]

Figure S2 (New addition)

Figure S2: Flowchart of experimental treatments to evaluate six treatment comparison.

Figure S3 (New addition)

Figure S3: Flowchart of experimental treatments to evaluate filtration by sterilized filter.

| | |
|---|---|
| Lines 177–182 | The $^{14}C$ concentrations and $\delta^{13}C$ values of two $NaHCO_3$-unfiltered samples, both before and after filtration assessments, were identical, indicating that the DIC of the $NaHCO_3$ solution itself remained unchanged during the assessment experiment (Fig. 1, Table S2). The $^{14}C$ concentrations of $NaHCO_3$-unfiltered, $NaHCO_3$-PES, and $NaHCO_3$-GF were consistent within the error range . Although each value was obtained from a single experiment, the five analytical results showed high consistency, indicating no detectable change in $^{14}C$ concentration due to filtration. This suggests that any increase in $^{14}C$ due to $CO_2$ contamination during filtration was not a significant concern. |
| Lines 259–261 | Since only a single time series of data is available for each treatment, the actual values for DIC changes remain uncertain. However, as the comparison is based on time series data, it can be posited that treatments exhibiting minimal change are highly effective. |

| Comment from reviewer | Answer |
|---|---|
| <RC1> Where appropriate, add 'groundwater' to the title and Abstract. The study tested both groundwater and seawater. | The title has been revised from "seawater" to "water sample" to encompass both seawater and groundwater. The result of groundwater was incorporated into the abstract. |
| <RC2> The authors presented a study to examine whether a bacteria inhibiting compound can be used in conjunction with filtration (at different pore sizes) in storing dissolved inorganic carbon samples for both isotope analysis of both seawater and groundwater (note groundwater needs to be included in the title). | We would like to modify the title to "water samples," which encompasses samples that are not limited to "seawater." |

| Revision for corresponding to above | |
|---|---|
| Title | Combining benzalkonium chloride addition with filtration to inhibit dissolved inorganic carbon alteration during the preservation of  water sample in radiocarbon analysis |
| Lines 10–11 | The present study aimed to evaluate the effectiveness of adding BAC as a disinfectant in carbon isotopic analyses of DIC in  water samples. |
| Lines 14–15 |  The freshwater sample that had undergone a BAC addition treatment showed the no alteration of DIC. In contrast, for seawater sample, BAC addition alone did not prevent changes in DIC, but the combined treatment was effective. |

| Comment from reviewer | Answer |
|---|---|
| <RC1> Line 16: Edit for clarity. Microbial activity was not directly measured. | As the reviewer commented, the degree of biological activity was not directly measured in this study. Instead, changes in dissolved inorganic carbon (DIC) were attributed to biological activity, and thus the expression "biological activity" was used in the text. However, this description is not entirely accurate and has been revised to "DIC changes" for clarity. |

| Revision for corresponding to above | |
|---|---|
| Lines 15–17 | The $^{14}C$ concentration of samples treated with both BAC addition and filtration exhibited minimal changes, ranging from 0.2–0.4 pMC over 41 weeks, despite the addition of sugar included to increase  DIC changes several-fold. |

| Comment from reviewer | Answer |
|---|---|
| <RC1> Line 83. Please add information on the salinity of the SW and GW to Table S1 and in the Abstract. The reader should be able to view this information to better understand what this text is talking about and/or more easily make an assessment whether it is appropriate for the SW sample to be called seawater, estuarine water, brackish water or something else. | The salinities of the SW and GW have been added to the abstract, the Materials and procedures section, and Table S1. The salinities of SW and GW were 20.4% and less than 0.05%, respectively. Therefore, SW is categorized as brackish water, and GW is categorized as freshwater. |
| <RC2> While there is nothing wrong with using metric units (mg/L) for solutes, as an analytical chemistry-oriented study, I would recommend using molar units (Table S1 needs to include salinity) to facilitate the comparison with seawater. | The description of chemical concentration has been changed to molar units in the revised manuscript and tables, and the salinity has been added to Table S1. |

| Revision for corresponding to above | |
|---|---|
| Lines 11–13 | We compared the efficacy of BAC addition, filtration (0.22 μm PTFE or 0.2–0.45 μm PES filters), and a combination of BAC addition and filtration in preventing DIC alterations caused by biological activity using the freshwater (salinity <0.05%) and the brackish water (salinity ~20%) samples. |
| Lines 108–111 | The salinities of the SW and GW were determined by summing of the chemical composition data to be 20.4% and below 0.05%, respectively (Table S1). While SW can be considered as a brackish water sample, but it is treated as a coastal seawater sample in the present study. For the SW, the expected values of DIC concentration is slightly smaller than 2 mmol·L$^{-1}$ and ~~$^{14}C$ concentrations were approximately 24 mg·L$^{-1}$ and ~100 pMC, respectively~~ that of $^{14}C$ concentration is ~100 pMC. |
| Table S1 | Table S1: Salinities and chemical compositions of SW and GW measured using the ion chromatography at the Geological Survey of Japan, AIST. The salinity values were obtained by summing the chemical composition data. |

| sample | Salinity (%) | Na$^+$ (mmol·L$^{-1}$) | K$^+$ (mmol·L$^{-1}$) | Mg$^{2+}$ (mmol·L$^{-1}$) | Ca$^{2+}$ (mmol·L$^{-1}$) | Cl$^-$ (mmol·L$^{-1}$) | SO$_4^{2-}$ (mmol·L$^{-1}$) |
|---|---|---|---|---|---|---|---|
| SW | 20.4 | 277.4 | 5.79 | 30.35 | 6.11 | 316.7 | 16.47 |
| GW | <0.05 | 0.41 | 0.14 | 0.27 | 0.43 | 0.25 | 0.009 |

| Lines 110, 112, 217–219 Tables 1, S1, S3 | mg·L$^{-1}$ ==> mmol·L$^{-1}$ (The values were changed to adapt unit.) |
|---|---|

| Comment from reviewer | Answer |
|---|---|
| <RC1> Figure 1, Figure 2, Table 1, and all supplementary Table captions should state what the data is (means?) and the number of replicates. | All values indicated in Figures 1 and 2 and Tables S2 and S3 were derived from a single measurement taken for each respective treatment. In contrast, the values shown in Table 1 represent the mean values of the respective measurements taken of the three preservation bottles for each period. The figure and table captions have been revised to clearly indicate whether the analytical values represent mean values or individual treatment sample values. |
| <RC2> For the information provided in Tables S2 and S3, it is unclear what the somewhat consistent uncertainty values are. Having a more detailed experimental setup will help to clear this up. | The description of uncertainty values, which are indicated to be 1σ of analytical error, will be added to the explanations in Tables S2 and S3. The standard deviations for $^{14}C$ measurements have been described into the "$^{14}C$ concentration and $\delta^{13}C$ measurements" section of the manuscript, and the method to obtain the error of $^{14}C$ measurement has been described as a standard calculation method in $^{14}C$ analysis with a citation. Since the description of $\delta^{13}C$ error had not been described, it has been incorporated into the "$^{14}C$ concentration and $\delta^{13}C$ measurements" section of the revised manuscript. |

| | |
|---|---|
| <RC2> Fig. 2 has no error bars, was each sample analyzed only once, what is the analytical precision and accuracy? | The error bars in Fig. 2 were not displayed, since the errors of $^{14}C$ and $\delta^{13}C$ are equivalent to or less than the size of the plotting symbols. This has been mentioned in the figure caption. The one plot was obtained from the single sample treatment. |
| <RC2> Table 1, show number of replicate analysis. | The explanation of Table 1 has been changed to show the number of replicate analysis: N=3. |

| Revision for corresponding to above | |
|---|---|
| Figure 1 | Figure 1: Comparisons of $^{14}C$ (a) and $\delta^{13}C$ (b) among the unfiltered and filtered solutions of 1 mmol·L$^{-1}$ of NaHCO$_3$. PES: filterd by PES disk filter (25 mm in diameter, 0.22 μm in pore size), GF: filtered by GF disk filter (25 mm in diameter, 1.0 μm in pore size). Each value of $^{14}C$ and $\delta^{13}C$ represents a single treatment occasion. The bars of $^{14}C$ concentration represent the measurement error of the AMS analysis. The $\delta^{13}C$ error is smaller than the size of the plotting symbols. |
| Figure 2 | Figure 2: Changes in $^{14}C$ and $\delta^{13}C$ during the preservation of SW and GW mixed with NaHCO$_3$ solution and beet sugar. Changes in DIC were augmented by the addition of beet sugar. (a) $^{14}C$ of SW, (b) $^{14}C$ of GW, (c) $\delta^{13}C$ of SW, (d) $\delta^{13}C$ of GW. The time series of DIC change for the respective treatments were derived from each experiment conducted on a single sample with varying preservation periods. The errors of $^{14}C$ and $\delta^{13}C$ are equivalent to or smaller than the size of the plotting symbols. |
| Table 1 | Table 1: Initial values of DIC concentrations and mean values of $\delta^{13}C$ with the standard deviation (1σ, N=3) for GW-Control2, GW-PES2 (0.2 μm), and GW-PES2 (0.45 μm), and the changes in $\delta^{13}C$ values with propagation error during the preservation. |
| Table S2 | Table S2: $^{14}C$ concentration and $\delta^{13}C$ value of NaHCO$_3$-unfilterd, NaHCO$_3$-PES and NaHCO$_3$-GF obtained from a single measurement each with the measurement errors (1σ). Due to the minimal changes in $^{14}C$ concentration, they are intentionally listed to two decimal places. |
| Table S3 | Table S3: DIC concentration, $^{14}C$ concentration, and $\delta^{13}C$ values for SW and GW obtained from a single occasion each. The errors of $^{14}C$ concentration and $\delta^{13}C$ values were represented in 1σ of the respective measurements. NA: not analyzed. |
| Lines 161–165 | The standard deviations for $^{14}C$ measurements were 0.02–0.04 pMC for waters with concentrations below 1 pMC, and less than 0.8% of $^{14}C$ concentration for waters above 10 pMC when measured at the Nagoya University. The precision of the quantitative analysis of carbon was better than 3%, and the background was below $2 \times 10^{-15}$ at KIGAM. The error of $^{14}C$ measurement was represented in 1σ based on a standard calculation method in $^{14}C$ analysis (Scott et al., 2007). |
| Lines 167–174 | The standard deviation of multiple $\delta^{13}C$ measurements by IRMS is less than 0.01‰, and for individual measurements, the error is represented as 1σ calculated from the variations in the dual inlet measurement. Some $\delta^{13}C$ measurements were performed using a continuous-flow IRMS coupled to a gas chromatography system (GC-IRMS; Delta-V Advantage with Gas Bench II, Thermo Fisher Scientific, Inc., USA) at the Geological Survey of Japan (Takahashi et al., 2019b). The standard deviation of multiple measurements of water samples by GC-IRMS is 0.04‰ (1σ). CO$_2$ gas was extracted from water by addition of phosphoric acid in a septum-sealed Exetainers® vial (12 mL, Labco Ltd., UK). The $\delta^{13}C$ value of each sample is represented as the averaged value over the three vials with the standard deviation (1σ). |

| Comment from reviewer | Answer |
|---|---|
| <RC1> Figure 1. There appear to be duplicate data points shown for some treatments (e.g. unfiltered, GF) but there is no explanation of what each represents. Are these replicates or different treatments? | The respective data points were obtained from a single treatment, therefore, the duplicate points for unfiltered and GF samples indicate two measurements from two individual treatments. The description of the experimental procedures corresponding to Figure 1 has been revised to understand the meaning of the respective data points. Figure 1 has been rearranged to show the order of filtration treatments using the NaHCO$_3$ solution. |

| Revision for corresponding to above | |
|---|---|
| Lines 93–98 | The filtration process was conducted on three occasions, once through a PES filter and twice through a GF filter, using the same NaHCO$_3$ solution, and the resulting three filtrates were obtained for each filtration treatment. Immediately after the respective filtrations, the  filtrates were injected into reaction containers to extract CO$_2$ from water sample. To ascertain whether any DIC was altered during the filtration experiment, one unfiltered NaHCO$_3$ solution each was introduced into the reaction containers before and after the three filtration treatments (Fig. S1).  |
| Figure 1 |
[Figure]
 Figure 1: Comparisons of $^{14}C$ (a) and $\delta^{13}C$ (b) among the unfiltered and filtered solutions of 1 mmol·L$^{-1}$ of NaHCO$_3$. PES: filterd by PES disk filter (25 mm in diameter, 0.22 μm in pore size), GF: filtered by GF disk filter (25 mm in diameter, 1.0 μm in pore size). Each value of $^{14}C$ and $\delta^{13}C$ represents a single treatment occasion. The bars of $^{14}C$ concentration represent the measurement error of the AMS analysis. The $\delta^{13}C$ error is smaller than the size of the plotting symbols. |

| Comment from reviewer | Answer |
|---|---|
| \<RC1\> Line 165. It is incorrect to conclude that filtration does not cause a change in 13C. There is not enough evidence for or against. Indeed, as there appears to be very low or no replication (n = 1 or 2) any differences in 13C among treatments may simply be an artefact. | As you correctly noted, the description that filtration does not cause a $\delta^{13}C$ change was not accurate and therefore has been deleted. Moreover, we concur with your comment that the observed differences in $\delta^{13}C$ among treatments may be attributed to an artefact. However, we maintain that the changes in $\delta^{13}C$ were not caused by atmospheric $CO_2$ contamination or degassing, and we propose that the observed $\delta^{13}C$ change may be attributed to an unidentified artifact factor other than filtration. This conclusion and a clarification stating that the effect of filtration on $^{14}C$ analysis is negligible have been added. |

| Revision for corresponding to above | |
|---|---|
| Lines 188–206 | These calculated values do not align with  the measured $^{14}C$ concentrations,  suggesting it can be confirmed that atmospheric $CO_2$ contamination does not occur through filtration. The $\delta^{13}C$ of DIC would change due to isotope fractionation associated with degassing. When DIC and gaseous $CO_2$ are in isotopic equilibrium, the $\delta^{13}C$ of DIC is typically higher than that of gaseous $CO_2$ (Zhang et al., 1995). As carbon with a lower $\delta^{13}C$ value is removed as $CO_2$ during degassing, the $\delta^{13}C$ of the remaining DIC in the solution would gradually increase. However, the measured $\delta^{13}C$ showed the opposite trend, indicating that the change in $\delta^{13}C$ is not due to $CO_2$ degassing from the $NaHCO_3$ solution. Thus, the two hypotheses—that $\delta^{13}C$ changes were caused by atmospheric $CO_2$ contamination or by degassing—were rejected. , and the observed $\delta^{13}C$ change may be attributed to an unidentified artifact factor other than filtration. Carbon contamination during sample treatments could significantly influence $^{14}C$ analysis, but the impact of isotopic fractionation can be eliminated by corrective calculations. Since the primary objective of this study is $^{14}C$ analysis, the effect of filtration is expected to be minimal or negligible. However, if $\delta^{13}C$ analysis were conducted, careful scrutiny and verification would be necessary. The experimental procedure in this study was designed to evaluate the maximum impact of filtration, and it is anticipated that the impact can be minimized by adopting experimental procedures that minimize filtration-related effects. |

| Comment from reviewer | Answer |
|---|---|
| \<RC1\> Line 186: "Beet sugar is more easily degraded". Compared to what? | The materials under comparison were BAC or other organic materials suspended in water. |

| Revision for corresponding to above | |
|---|---|
| Lines 225–227 | It is reasonable to assume that these large changes in $^{14}C$ concentration and $\delta^{13}C$ were caused by the DIC derived from beet sugar, given that beet sugar is more easily degraded than BAC or other organic materials suspended in water. |

| Comment from reviewer | Answer |
|---|---|
| \<RC1\> Line 204: " …reduced changes in DIC….". Compared to the control treatment? | Yes. It is compared to the Control sample. |

| Revision for corresponding to above | |
|---|---|
| Lines 249–250 | Filtration, BAC addition, and the combined treatment  indicated more effective than the Control samples in preserving DIC in water samples (Fig. 2). |

| Comment from reviewer | Answer |
|---|---|
| \<RC1\> Line 210-211: Edit for clarity.
 Line 213: This is not true. Filtration + BAC essentially stopped changes in DIC for both GW and SW. | The description was inaccurate and has been deleted. |
| \<RC2\> Line 213-217, this seems quite wordy as it's already in the table | To streamline the text, the manuscript has been changed to delete isotopic changes, but to remain the sample name. |

| Revision for corresponding to above | |
|---|---|
| Lines 253–259 | This confirms that BAC alone is not suitable for seawater samples and indicates that the seawater sample utilized in this investigation contains constituents, probably unique microorganisms, whose biological activity cannot be entirely suppressed by BAC addition, as reported in previous studies.  Accordingly, SW represents an appropriate sample for the purpose of investigating potential methods for addressing the issue of BAC impairment in seawater. The $^{14}C$ concentrations and $\delta^{13}C$ values were observed to be relatively constant in samples treated with both filtration and BAC: SW-PTFE+BAC, SW-PES+BAC, GW-BAC, GW-PTFE+BAC, and GW-PES+BAC. The $^{14}C$ and $\delta^{13}C$ changes were minimal SW-PTFE+BAC,  SW-PES+BAC,  GW-BAC,  GW-PTFE+BAC, and  for GW-PES+BAC (Table S3). |

| Comment from reviewer | Answer |
|---|---|
| \<RC1\> Line 36. Remove "have been" and ". They "
 Line 63. Replace "is" with 'are'
 Line 273: Remove full stop after "However" | The manuscript has been revised in accordance with all technical recommendations. |

| Revision for corresponding to above | |
|---|---|
| Lines 36–37 | The methods  proposed for the preservation of water samples without the use of $HgCl_2$.  include refrigeration, filtration, and the addition of non-toxic or less toxic preservatives |

| Lines 64–65 | Gloël et al. (2015) noted that the factor diminishes the effectiveness of BAC in seawater over time is likely spores,  which are resistant to heat and sterilization. |
|---|---|
| Lines 316–317 | However degradation of BAC, a refractory organic compound, only begins after readily decomposable organic matter is fully consumed |

| Comment from reviewer | Answer |
|---|---|
| \<RC2\> Even though DIC concentration isn't a focus of this study but isotopes, it would be helpful to present the DIC concentration changes through time in these different treatments along with analytical uncertainty, if such information is available. | Unfortunately, due to limitations in the analytical equipment, we were unable to measure the DIC concentration. It is possible to calculate it from the weight of the water sample used for $CO_2$ extraction and the amount of $CO_2$ collected. However, this approach did not provide sufficient precision to distinguish between subtle DIC changes. |

| Comment from reviewer | Answer |
|---|---|
| \<RC2\> Line 101, ReCEIT needs to be briefly explained in addition to the citation. | The explanation of the ReCEIT has been added. |
| Revision for corresponding to above | |
| Lines 149–155 | $CO_2$ extraction from water samples for the measurement of $^{14}C$ concentration and $\delta^{13}C$ values was performed using the ReCEIT (repeated cycles of extraction, introduction, and trapping) method (Takahashi et al., 2021), which is a simple and carrier gas-free method that can handle a variety of water samples with a wide range of DIC concentrations (0.4–100 mmol/L, in the case of 1 mmol of carbon) and produces high $CO_2$ yields. The procedure is composed of repeating the cycles of $CO_2$ extraction from water into the headspace of the reaction container, expansion of the extracted gas into the vacuum line, and cryogenic trapping of $CO_2$. This method, which extracts $CO_2$ without bubbling, is particularly well-suited for BAC-added samples, which tend to foam. |

| Comment from reviewer | Answer |
|---|---|
| \<RC2\> Line 191, remove "salt". | The description has been changed. |
| Revision for corresponding to above | |
| Lines 232–233 | It can be reasonably assumed that water discharged from tidal flats will have higher concentrations of organic carbon,  nutrients, and microbes than typical seawater |

| Comment from reviewer | Answer |
|---|---|
| \<RC2\> Line 192-195, explain what the "boost effect" is. | The explanation of the boost effect has been added above the mentioned lines. |
| Revision for corresponding to above | |
| Lines 227–231 | Given that DIC change during the preservation was enhanced by the incorporation of sugar, it is imperative to ascertain the impact of sugar addition. Takahashi and Minami (2022) defined the boost effect as an index of how many times the DIC change in the sugar-added sample is greater than the DIC concentration change in the no-sugar sample during the preservation. It is anticipated that the boost effect will be more pronounced in instance where there is a paucity of organic matter and a greater prevalence of microorganisms in the water sample. |

---

## Referee Report (RR1)

The authors present another insightful study on the preservation of water samples for radiocarbon and dissolved inorganic carbon (DIC) analysis, employing methods that avoid the use of toxic preservatives such as mercury. This work is particularly significant as it demonstrates that sample integrity can be maintained for up to 285 days for seawater and 126 days for groundwater using a combination of filtration and benzalkonium chloride (BAC) treatment. These extended preservation times represent a clear advancement over the authors' previous findings.

Overall comment: The newly added supplemental flowcharts effectively clarify the study's methodology. To further enhance usability, it would be beneficial to include a final summary flowchart—ideally in the form of a decision tree—to guide researchers in selecting the most appropriate preservation method for either groundwater or seawater samples.

Minor adjustments:

Line 14: The freshwater sample treated with  BAC  showed  no alteration of DIC.

Line 24: For global understanding of ocean water behaviors, it is necessary to analyze  samples from various regions over long timeframes, . . .

Line 67: possibly due to interaction with  components in the seawater,

Line 109: While SW can be considered as a brackish water sample,  it is treated as a coastal seawater sample in this study.

Line 188: These calculated values do not align with the measured 14C concentrations, suggesting  that atmospheric CO2 contamination **is unlikely**.

Line 230: It is anticipated that the boost effect will be more pronounced in instance**s** where . . .

Line 231 - 232: The SW in this study was sampled at a tidal flat location  . .

Line 251: the results were consistent with  previous studies

Line 287: this $\delta$ 13C change seems to be negligible, given its small magnitude and the associated uncertainty which  only became detectable through sugar-induced microbial activity magnification.. . .

---

## Author Response (AR2)

We sincerely thank the anonymous referees for their time and thoughtful feedback on our manuscript. We have revised the manuscript according to their comments. Responses to each comment are provided below. We believe that we have addressed all of the comments. In the table below, additions to the text are highlighted in red, and deletions are in highlighted blue with strikethrough. Line numbers indicate those in the revised manuscript.

| Comment from Anonymous referee #2 | Answer and revision |
|---|---|
| 1. Salinity has no unit in typical ocean science papers. I also understand you meant ‰, not %. Regardless, please remote the symbol in both abstract and Line 108-109. | The descriptions of salinity Line 108 and Table S1 are now expressed in ‰ and units have been removed. |
| 2. L112, pMC should be defined in Line 110. | I missed the pMC on line 84. Since the first description in the manuscript is here, the article for the definition in pMC has been cited on line 84-85. Line 111 has not been revised, and the citation on line 112 has been deleted.
Since the pMC was described without the definition in the abstract (Line 16), "percent Modern Carbon" has been added. |
| 3. Line 188-189, the sentence is incomplete. | The description has been revised.
Line 187-188: These calculated values do not align with the measured $^{14}C$ concentrations, suggesting  that atmospheric $CO_2$ contamination is unlikely. |
| 4. Line 230, just "corrections", but please specify what type of correction. | I could not find the description of the "correction" in line 230. I realized you meant the correction on line 159. I cited an article that described the isotope correction.
Line 159-160: **Corrections** for isotopic fractionation (Stuiver and Polach, 1977) were performed using the $^{13}C/^{12}C$ ratio measured by AMS. |
| 5. Change all concentration units to mM, (mmol L-1 is not standard usage). | The Mathematical notation and terminology in the Submission Guidelines of OS defines that the unit is expressed using the SI unit. The mole is expressed as "mol" in the SI unit. On the other hand, the liter (L) is not SI unit, but it is a non-SI unit accepted for use with the SI. Therefore, we have chosen to use $mol \cdot L^{-1}$, which is not an SI unit, because it is more familiar and commonly used in our field. For this reason, we do not agree with this comment and have not made any changes to the description. |
| 6. After examining Fig 2, I would recommend using time as x-axis and plot each treatment separately, even though some of the data points may overlap. | Figure 2 has been revised to display preservation days on the x-axis. To avoid overlapping plots, the zero point on the x-axis has been offset for each treatment. |

| Comment from Anonymous referee #3 | Answer and revision |
|---|---|
| Overall comment: The newly added supplemental flowcharts effectively clarify the study's methodology. To further enhance usability, it would be beneficial to include a final summary flowchart—ideally in the form of a decision tree—to guide researchers in selecting the most appropriate preservation method for either groundwater or seawater samples. | Summary flowchart has been added as Figure 3 in Conclusions. |
| Line 14: The freshwater sample treated with  BAC  showed  no alteration of DIC. | The description has been revised.
Line 14: The freshwater sample treated with  BAC  showed the no alteration of DIC. |
| Line 24: For global understanding of ocean water behaviors, it is necessary to analyze  samples from various regions over long timeframes, . . . | The description has been revised.
Line 25: For global understanding of ocean water behaviors, it  is necessary to analyze samples from various regions over long timeframes |

| | |
|---|---|
| Line 67: possibly due to interaction with  components in the seawater, | The description has been revised.
Line 66-67: possibly due to interaction with  components in the seawater, |
| Line 109: While SW can be considered as a brackish water sample,  it is treated as a coastal seawater sample in this study. | The description has been revised.
Line 109-110: While SW can be considered as a brackish water sample,  it is treated as a coastal seawater sample in this study. |
| Line 188: These calculated values do not align with the measured 14C concentrations, suggesting  that atmospheric CO2 contamination **is unlikely**. | The description has been revised.
Line 187-188: These calculated values do not align with the measured $^{14}C$ concentrations, suggesting  that atmospheric $CO_2$ contamination is unlikely. |
| Line 230: It is anticipated that the boost effect will be more pronounced in instance where . . . | The description has been revised.
Line 229: It is anticipated that the boost effect will be more pronounced in instances where |
| Line 231 - 232: The SW in this study was sampled at a tidal flat location  . . | The description has been revised.
Line 230-231: The SW in this study was sampled  at  a tidal flat location along the Pacific coast, near the estuaries of major rivers. |
| Line 251: the results were consistent with  previous studies | The description has been revised.
Line 248: the results were consistent with  previous studies |
| Line 287: this δ13C change seems to be negligible, given its small magnitude and the associated uncertainty which  only became detectable through sugar-induced microbial activity magnification.. . . | The description has been revised.
Line 283-285: this $\delta^{13}C$ change seems to be negligible given its small magnitude and  uncertainty  which only became detectable through sugar-induced microbial activity magnification. |

---

## Author Response (AR3)

We sincerely thank the anonymous reviewers and the editor for their thoughtful consideration in the publication of this paper.

The errors in the salinity descriptions have been corrected. The correct salinity values are <0.05 and 20.4. In addition to the error of 204 on line 108, we have found and corrected other errors. These corrections appear on lines 14 and 108 in the manuscript and in Table S1.

| | before correction | after correction |
|---|---|---|
| Line 14 | biological activity using the freshwater (salinity <0.5) and the brackish water (salinity ~200) samples. | biological activity using the freshwater (salinity <0.05) and the brackish water (salinity ~20) samples. |
| Line 108 | summing of the chemical composition data to be 204 and below 0.5 | summing of the chemical composition data to be 20.4 and <0.05 |
| Table S1 | 204 and <0.5 in the salinity column | 20.4 and <0.05 in the salinity column |